# Media Pressures, Internalization of Appearance Ideals and Disordered Eating among Adolescent Girls and Boys: Testing the Moderating Role of Body Appreciation

**DOI:** 10.3390/nu14112227

**Published:** 2022-05-26

**Authors:** Rasa Jankauskiene, Migle Baceviciene

**Affiliations:** 1Institute of Sport Science and Innovations, Lithuanian Sports University, 44221 Kaunas, Lithuania; rasa.jankauskiene@lsu.lt; 2Department of Physical and Social Education, Lithuanian Sports University, 44221 Kaunas, Lithuania

**Keywords:** positive body image, internalization of appearance ideals, media use, media pressures, disordered eating, adolescents

## Abstract

The aim of the present study was to test the moderating role of body appreciation in the mediation model of media pressures, internalization of appearance ideals, and disordered eating in adolescents. One thousand four hundred and twelve Lithuanian adolescents (40.4% were boys, age range: 15–18 years) participated in the cross-sectional study. The mean age of the sample was 16.9 (SD (standard deviation) = 0.5) for girls and 17.0 (SD = 0.4) for boys. Adolescents completed a questionnaire consisting of measures of body appreciation (Body Appreciation Scale-2 (BAS-2)), disordered eating (Eating Disorder Examination Questionnaire 6 (EDE-Q 6)), attitudes towards sociocultural pressures towards appearance (Sociocultural Attitudes towards Appearance Questionnaire 4 (SATAQ-4)), and time browsing internet for leisure purposes. To assess the primary hypotheses, moderated mediation models were tested separately in boys and girls. In girls, body appreciation moderated associations between media pressures and internalization of thin body ideals and between internalization of thin appearance ideals and disordered eating. In boys, body appreciation moderated only the association between media pressures and disordered eating. The effect of media pressures on disordered eating was the highest in boys with the lowest body appreciation. Body appreciation protects adolescent girls from internalization of thin appearance ideals in the presence of media pressures and from disordered eating in the case of internalization of thin body appearance ideals. In boys, body appreciation provides a protective effect from media pressures towards appearance to disordered eating behaviors. The findings of the present study could inform intervention programs that aim to prevent disordered eating, strengthen positive body image, and promote healthy eating in adolescent girls and boys. Specific programs might be beneficial in preventing disordered eating in boys with low body appreciation.

## 1. Introduction

Adolescence is a period of identity formation, self-expression, and peer acceptance. Media use is an important agent for online and offline developmental issues of adolescents [1]. The use of traditional and especially social media is abnormally popular among adolescents [2,3,4,5,6]. Evidence from neuroscience suggests that adolescents are highly sensitive to acceptance and rejection of media, and their heightened emotional sensitivity and protracted development of reflecting processing and cognitive control may make them especially reactive to emotion-arousing media [7]. Adolescent traditional and social media use was associated with worse sleep quality, lower life satisfaction and self-esteem, psychological distress, depression, anxiety, lower physical activity, lower body image, body shame, and disordered eating [4,8,9,10,11,12,13,14].

According to the sociocultural theory of eating disorders, namely the Tripartite influence model (TIM) [15,16,17], media pressures are one of the most important agents affecting adolescent disordered eating through the mediators such as internalization of appearance ideals, social comparison, and body dissatisfaction [18,19]. The media promotes appearance ideals that are stereotyped and unrealistic [20,21] and refers to thin and/or fit body image for women and muscular bodies with low body fat for men [22,23]. Internalization of appearance ideals is a multifaceted construct that encompasses the cognitive-affective acceptance of and desire to conform to a societal standard, the cognitive and affective integration of the value and meaning of that standard into a personal value system, and compliance with or engagement in behavior in line with this standard [24,25].

While media pressures are prevalent, the extent to which appearance ideals are internalized might be different between adolescents. Recent evidence suggests that high self-esteem and healthy identity formation protect adolescents against appearance ideals internalization [26,27,28]. Therefore, it is important to understand what other individual factors might protect adolescents from internalization of appearance ideals, body dissatisfaction, and disordered eating. One of the possible moderators might be positive body image [29].

Positive body image represents body appreciation, body pride, and body acceptance [29]. The central facet of positive body image is body appreciation defined as accepting, holding favorable opinions towards, and respecting the body; resisting the sociocultural pressures to internalize stereotyped beauty standards as the only form of human beauty; and appreciating the functionality and unique aspects of the body [30,31]. Studies in adolescents have demonstrated that positive body image was associated with greater self-esteem, body satisfaction, body functionality, less favorable attitudes towards cosmetic surgery, sexual objectification, less internalization of appearance ideals, and higher intuitive eating [32,33,34,35,36,37,38].

Adolescents with high body appreciation might be less vulnerable to media pressures towards appearance since they might more strongly reject the stereotyped beauty ideals usually exposed in traditional and highly visual media [29]. Further, adolescents high in body appreciation might demonstrate lower internalization of beauty ideals since they have an attitude that beauty takes a variety of forms and is not limited to stereotyped thin, fit, or muscular ideals. A recent study in Chinese adolescents showed that body appreciation moderated the indirect associations between selfie viewing and adolescent facial dissatisfaction via general attractiveness internalization [39]. Therefore, it is reasonable to expect that adolescent body appreciation will moderate the association between media pressures, internalization of appearance ideals, and disordered eating.

Adolescents with high body appreciation and internalizing appearance ideals might still be satisfied with their bodies because they respect and accept their body and its unique features and therefore might demonstrate more positive attitudes towards eating [29]. Previous studies on adult women have shown that women who reported high body appreciation expressed less negative exposure effects from thin appearance ideals [40,41]. Nevertheless, one experimental study found that exposure to images of women in active, fitness contexts produced appearance and functionality dissatisfaction in women, and no moderation effects of body appreciation were observed [42]. Therefore, it is important to understand the role of positive body image in associations between media pressures, internalization of thin body ideals, and disordered eating, especially in adolescents. Adolescents are active media users and are highly underrepresented in research on positive body image. Further, it remains unknown whether positive body image might be protective of adolescent boys’ eating behaviors. Research in adolescent boys’ body image is still scarce. It remains unknown whether positive body image and relevant psychological mechanisms that apply to adolescent girls would also be found in boys.

### The Present Study

The aim of the present study was to test the moderating role of positive body image (operating as body appreciation) in the mediation models of media pressures, internalization of appearance ideals, and disordered eating in adolescent girls and boys. For this purpose, we developed a hypothesized model of moderated mediation (Figure 1).

In the present study, we expected that body appreciation would moderate the associations between mediation models of girls and boys. In other words, we expected that associations between media pressures, internalization of thin and low body fat ideals, and disordered eating would be weaker in girls and boys expressing more positive body image and vice versa.

## 2. Materials and Methods

### 2.1. Procedure and Study Participants

First, permission to conduct the study was received from the Lithuanian Sports University Social Research Ethics Committee (Protocol No. SMTEK-32, 27 September 2019). Next, permission to conduct the study was obtained from schools’ directors or principals. With the help of class teachers in each gymnasium, information about the study and study aims was spread to parents or caregivers, asking for their oral consent for schoolchildren to participate in the study.

Study participants had a possibility to agree or to disagree to participate in the survey by themselves, selecting an appropriate option in the online form: “I agree to participate” or “I disagree to participate”. Only agreeing students were provided with the study measures. In addition, study participants could close the online form at any point without recording their answers.

Data were collected using an online survey form (Google Forms) and conducted in late 2019 before the pandemic arose: from November 2019 to January 2020. Forty-one schools from 26 Lithuanian cities and towns participated in the study, representing the majority of country municipalities. All adolescents involved in the study were from 11th grade with an age range of 15–18 years, 92% of which were 17 years old. The mean age of participant boys was 17.0 ± 0.4 years and 16.9 ± 0.5 years for girls. In total, 1492 students completed the survey form. Out of them, 56 refused to participate in the study. The reasons for refusal were not collected. In addition, 24 records were deleted because they were completed incorrectly (for example, the actual and desired weight or height indicated were not realistic). The final sample consisted of 1412 students (59.6% were girls, 40.4% were boys), containing no missing data.

### 2.2. Measures

Sociodemographic variables included information on gender, age, city (town), and name of a gymnasium. In addition, self-reported weight and height were used to calculate body mass index (BMI). In this sample, mean BMI was 22.0 ± 3.0 kg/m^2^ (range: 15.4–40.6 kg/m^2^) in boys and 21.0 ± 3.1 kg/m^2^ (range: 14.0–41.7 kg/m^2^) in girls. Using the International Obesity Task Force (IOTF) BMI cutoffs [43], all study participants were classified into underweight (4.9% of boys and 16.5% of girls, respectively), normal weight (79.1 and 72.9%), and overweight/obese (16.0 and 10.6%).

Time browsing the internet was assessed using a single question asking participants to indicate the average duration of hours per day of browsing the internet for noneducational purposes, communicating with peers on social media, watching videos, and listening to music.

Internalization of a thin and muscular/athletic body was taken from the Sociocultural Attitudes Towards Appearance Questionnaire 4 (SATAQ-4) [25]. This measure demonstrated good psychometric properties, and the original scale structure was confirmed in a large sample of Lithuanian students of both genders [44]. The SATAQ-4 questionnaire consists of five subscales: Internalization of Thin/Low Body Fat; Internalization of Muscular/Athletic Body; Pressures from Family; Pressures from Media; Pressures from Peers. Internalization of Thin/Low Body Fat comprises five statements and allows assessing the degree to which individuals endorse or strive towards thin/low body fat ideals. Examples of items: “I think a lot about looking thin”; “I want my body to look like it has little fat”. The respondents provided five response options on a Likert scale from 1—disagree up to 5—agree. Before the analysis, response options were averaged with the higher score, indicating higher internalization of thin/low body fat ideals. For this study, Cronbach’s α for internalization of the thin/low body fat subscale was 0.94 for girls and 0.89 for boys.

Perceived pressures towards appearance from the media subscale comprise four statements (sample item: “I feel pressure from the media to improve my appearance”), and the final averaged score in the range from 1 up to 5 demonstrates higher pressure towards appearance from media (magazines, movies, internet, advertisements). In this study, internal consistency of the media pressures subscale was excellent (Cronbach’s α: 0.97 for girls and 0.92 for boys).

Disordered eating was assessed using the Eating Disorder Examination Questionnaire 6.0 (EDE-Q 6.0) [45]. The EDE-Q 6.0 helps to comprehensively evaluate the essential behavioral and attitudinal characteristics of disordered eating behavior and eating disorders. It consists of 28 items. Six open-ended questions assess the frequency data on the essential behavioral characteristics of disordered eating: binge eating, self-induced vomiting, laxative use, and excessive exercise. A further 22 attitudinal questions consisting of four subscales reflect the severity of disordered eating characteristics. The answers are varied on a six-point Likert scale from 0 (no day) to 6 (every day), and answers are averaged. A higher score indicates either greater severity or frequency of disordered eating. In the present study, we used the general score of disordered eating. The Lithuanian version of the scale demonstrated good psychometric properties [46]. In the present study, internal consistency for the general scale was good: 0.92 for boys and 0.95 for girls.

The Body Appreciation Scale-2 (BAS-2) was used to assess positive body image [30]. A unidimensional ten-item scale is designed to measure body acceptance, body appreciation, and resistance to pressures towards appearance. Sample item of BAS-2: “My behavior reveals my positive attitude toward my body; for example, I hold my head high and smile”. Response options are provided on a Likert scale from “never” up to “always”. A final averaged score of the response options indicates a higher level of body appreciation. Unidimensional factor structure and good psychometric properties were confirmed in the Lithuanian translation of the BAS-2 in adolescents [33]. For this study, Cronbach’s α was 0.97 for both boys and girls.

### 2.3. Statistical Analysis

Using the continuously varying sample size approach to Monte Carlo power analysis, approximately 150 individuals are required to ensure a statistical power of at least 80% for detecting the hypothesized indirect effect [47]. For the simple model with one mediator, a power of 0.80 can be achieved by a sample size of 50–200. The calculated power for the sample size *n* = 842 (girls) and *n* = 570 (boys) was 1.00.

First, a series of independent-samples t-tests were conducted to examine any gender differences among variables, and effect sizes with Hedges’ g correction for different sample sizes were calculated. Effect sizes above 0.2 were considered small, equal or above 0.5 were moderate, and equal or above 0.8 were strong [48]. Bivariate correlations between study variables were tested by Pearson correlations. Internal consistency of the scales was tested by Cronbach’s α. Magnitudes between 0.1 and 0.3 were considered small, above 0.3 and below 0.5 were moderate, and equal or above 0.5 were strong with a significance level of <0.05 [49]. Preliminary statistical analysis was carried out using IBM SPSS Statistics v27 (IBM Corp., Armonk, NY, USA).

Next, to assess the primary hypotheses, a moderated mediation model was tested using the Mplus v8.7 (Muthén & Muthén, Los Angeles, CA, USA). Specifically, the hypothesized model predicted that the association between media pressures and disordered eating will be mediated by internalization of thin/low body fat ideals and that all the associations would be moderated by body appreciation in boys and girls (see Figure 1). All statistical tests were two-tailed, and significance was determined at the 0.05 level. A bootstrapping procedure was used to test the significance of the total and indirect effects and the differences in these effects across levels of the moderator variables with 5000 bootstrap samples [50,51]. The 95% confidence intervals for the coefficients calculated by bootstrapping methods were considered statistically significant if the confidence intervals did not include zero. Codes for data visualization were generated to represent the conditional effects of the focal predictors and were conducted with the SPSS syntax file. All of the Mplus codes for moderated mediation models were obtained from the website (accessed on 3 April 2022, available on: http://www.offbeat.group.shef.ac.uk/FIO/mplusmedmod.htm#modindex) [51].

## 3. Results

Bivariate correlations and comparison of the study measures calculated as means and standard deviations for each variable are presented in Table 1 and Table 2. Results indicated that there were statistically significant gender differences in all study variables with small to medium effect sizes. Mean hours of browsing the internet for noneducational purposes per day, perceived media pressures, disordered eating, and internalization of thin and low body fat ideals were higher in girls. Mean body mass index and body appreciation scores were higher in boys.

Time spent browsing the internet for noneducational purposes was weakly correlated with disordered eating in boys and girls. In addition, a longer duration of internet use correlated with thin and low body fat internalization in boys, and there was a slight negative correlation with body appreciation in girls. Moreover, the time of browsing the internet was correlated with media pressures neither in boys nor in girls. Higher body mass index was correlated with a higher level of media pressures, thin body internalization, disordered eating, and lower body appreciation. There were positive correlations between media pressures, thin body internalization, and disordered eating in both gender groups. In addition, there were negative moderate correlations between body appreciation and disordered eating in both gender groups. All correlations observed were stronger in girls as compared to boys. In addition, boys’ internalization of thin body ideals had no association with body appreciation.

The final model of moderated mediation in girls with standardized regression coefficients is presented in Figure 2. The final model represents significant path coefficients from media pressures to internalization of thin body ideals (B (estimate) = 0.60, 95% CI (0.46–0.73)), from internalization of thin body to disordered eating (B = 0.59, 95% CI (0.42–0.77)), and from media pressures to disordered eating behaviors (B = 0.36, 95% CI (0.20–0.53)). Moreover, a mediated effect from media pressures to disordered eating was also established: B = 0.35, 95% CI (0.23–0.50), total effect: B = 0.72, 95% CI (0.57–0.86). Comments provide nonstandardized path coefficients with the 95% CIs, while Figure 2 standardizes path coefficients.

Next, in the final model, there were two significant moderating effects of body appreciation: from the association between perceived media pressures to internalization of the thin appearance ideals and from internalization of thin body ideals to disordered eating. Specifically, examination of conditional effects in different levels of moderators revealed that among girls, the effect of media pressures on internalization of thin body ideals was significant at each level of body appreciation, but the effect was the highest in the group of the lowest positive body image: low, B = 0.43, 95% CI (0.36–0.50); average, B = 0.34, 95% CI (0.28–0.39); and high, B = 0.25, 95% CI (0.16–0.33). Similar results were obtained by examining the moderating effect of body appreciation in the association between internalization of thin body ideals and disordered eating. The conditional effects in different levels of moderator were as follows: low, B = 0.25, 95% CI (0.18–0.32); average, B = 0.19, 95% CI (0.14–0.24); and high, B = 0.13, 95% CI (0.06–0.20). Finally, conditional indirect effects from media pressures to disordered eating in the levels of moderators were as follows: low, B = 0.19, 95% CI (0.14–0.25); average, B = 0.13, 95% CI (0.10–0.16); and high, B = 0.08, 95% CI (0.05–0.11). The moderating effect of body appreciation in the association between media pressures and disordered eating was not significant. Visualizations of the body appreciation moderating effects are presented in Figure 3 and Figure 4.

Similar results were obtained when testing the mediating effect of internalization of thin body ideals with low body fat in boys (Figure 5). Significant path coefficients from media pressures to internalization of thin body ideals (B = 0.38, 95% CI (0.29–0.47)), from internalization of thin body to disordered eating (B = 0.26, 95% CI (0.20–0.32)), and from media pressures to disordered eating behaviors (B = 0.55, 95% CI (0.33–0.80)) were found. In addition, a mediated effect from media pressures to disordered eating was also established: B = 0.10, 95% CI (0.07–0.13), total effect: B = 0.64, 95% CI (0.43–0.90).

Next, in the boys’ model, there was only one significant moderating effect of body appreciation in the association between perceived media pressures towards appearance to disordered eating. Examination of conditional effects in different levels of moderators revealed that among boys, the effect of media pressures on disordered eating was significant only in the group with the lowest (B = 0.30, 95% CI (0.20–0.40)) and intermediate (B = 0.17, 95% CI (0.10–0.25)) body appreciation level, represented by the sample mean of the BAS-2. In boys with the highest level of BAS-2 represented as plus 1 SD to the sample mean, the effect of media pressures on disordered eating was not significant (B = 0.05, 95% CI (−0.06–0.16)). Visualization of the body appreciation moderating effect is presented in Figure 6.

## 4. Discussion

Our study provides important new data suggesting that body appreciation is a moderator in a mediation model of media pressures, internalization of beauty ideals, and disordered eating in adolescent girls. In boys, body appreciation moderates the association between media pressures and disordered eating. This is a novel contribution to science and practice, since our study adds novel empirical data informing disordered eating prevention programs. These data are also important for programs that aim to prevent overweight and obesity and promote healthy eating in adolescents.

Our results suggested that a protective role of body appreciation in girls was observed at each of its levels. The lowest effect of media pressures on internalization of thin body ideals was observed in girls with the highest body appreciation. The same protective role was observed in the association between internalization of thin body ideals and disordered eating. These results are in accordance with the conception of positive body image and previous research suggesting that adolescent body appreciation is associated with higher resistance to sociocultural pressures towards appearance, less internalization of beauty ideals, and more positive eating attitudes and behaviors [29,31,36]. The findings of the present study suggest that disordered eating prevention programs might benefit from including teaching about positive body image. Adolescents spend a major part of their leisure time using media and experience media pressures daily. This study adds new empirical data suggesting that positive body image might protect adolescent girls from the detrimental effect of internalization of stereotyped thin body ideals in the presence of media pressures and prevent disordered eating in the presence of internalization of thin body ideals.

Our study has an important contribution to expanding knowledge about the role of positive body image in boys. The results of the present study suggested that body appreciation does not moderate associations between media pressures and internalization of thin body ideals and between the later and disordered eating. Possibly, the results for girls and boys were different since we tested internalization of thin body ideals as a mediator between media pressures and disordered eating. Typically, boys and men internalize muscular and athletic body types with low body fat but not thin body ideals that are more relevant for girls [23]. Therefore, future studies should test a partial model with internalization of muscular or athletic body ideals as a mediator between media pressures and disordered eating. However, in the present study, the moderating effect of positive body image in boys was observed in direct associations between media pressures and disordered eating. Important new findings of this study showed that the most damaging effect of media pressures on disordered eating was observed in boys with low body appreciation. In other words, our results suggest that high body appreciation might be less important for adolescent boys’ eating attitudes and behaviors compared to girls, but low body appreciation is detrimental since, in boys with low body appreciation, the effect of media pressures on disordered eating is the highest. These results inform disordered eating prevention programs for adolescent boys. Universal disordered eating prevention programs might benefit from strengthening the positive body image of boys. Additionally, specific interventions for boys with low body appreciation might be effective in preventing disordered eating.

In the present study, adolescent boys reported higher body appreciation compared to girls, and these findings are in line with findings from previous studies [52]. Disordered eating, media pressures, and internalization of thin appearance ideals reflect previous findings and gender tendencies on stereotyped beauty ideals for women and men [26,53]. Finally, our study showed that adolescents spent approximately four hours daily browsing the internet in their leisure time. Importantly, girls spent significantly more time on the internet compared to boys. This might be explained by findings that girls spend more time on social networks [4,54]. Time spent using the internet was not associated with media pressures. However, we assessed time spent browsing the internet in leisure time, but not specifically time spent on digital news and/or social media, as well as the intensity and involvement in these activities. Recent studies showed that the magnitude of associations between media use and body-image-related outcomes significantly varied across media types, extending to the use [11,21,55] and content of media [56]. Future studies should address this issue. Finally, time spent browsing the internet was weakly but significantly associated with adolescents’ disordered eating. These findings are in line with the conclusions of the previous research [13].

This study benefited from some strengths. First, the sample size of adolescents was large, all measures were psychometrically sound [57], and the data were analyzed using advanced statistical techniques. This study presents several novel aspects, including testing body appreciation as an outcome of interest for boys. However, despite its strengths, the present study has important limitations that are worth consideration. First, the present study is cross-sectional in nature, and causal inferences could not be assumed. It is important that future research tests our assumptions longitudinally. Additionally, the present study only considered the mediating role of internalization of thin body ideals, and future studies should test muscular athletic internalization. Finally, body appreciation is one of the main facets of positive body image; however, other important aspects such as body functionality should also be evaluated in future studies since they might also moderate mediated associations in models of media pressures, internalization of beauty ideals, and disordered eating [58].

## 5. Conclusions

The findings of the present study suggest that body appreciation is a moderator in a mediation model of media pressures, internalization of beauty ideals, and disordered eating in adolescent girls. In boys, body appreciation moderated the association between media pressures and disordered eating. The weakest associations between media pressures, internalization of thin body ideals, and disordered eating were observed in girls with the highest body appreciation. The effect of media pressures on disordered eating was the highest in boys with low body appreciation. The findings of the present study inform universal intervention programs that aim to prevent disordered eating and promote healthy eating habits. Specific programs might be beneficial in preventing disordered eating in boys with low body appreciation.

## Figures and Tables

**Figure 1 nutrients-14-02227-f001:**
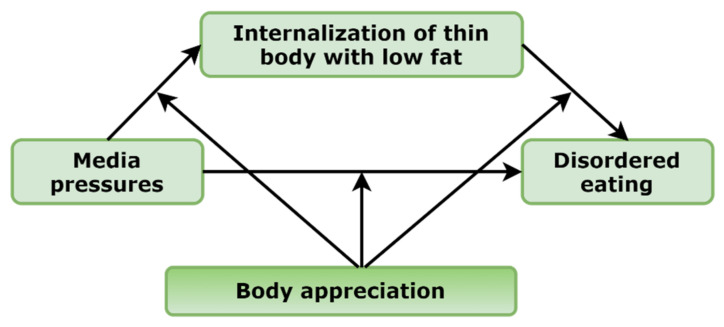
Hypothetical model of moderated mediation.

**Figure 2 nutrients-14-02227-f002:**
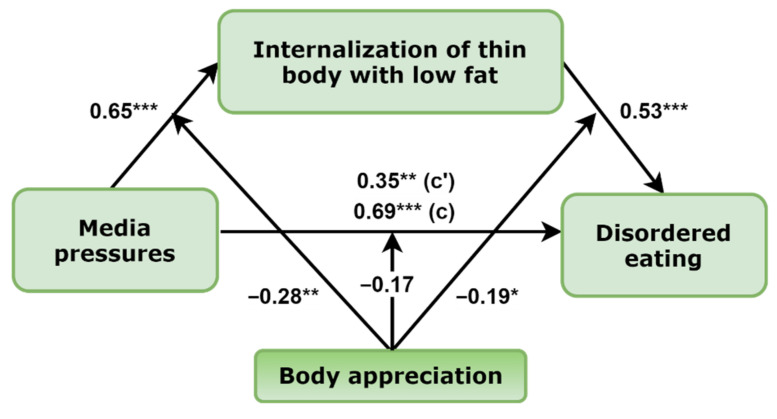
The final moderated mediation model in girls with the mediating effect of internalization of thin body (*n* = 842). Note: standardized path coefficients are presented; c’ = direct effect, c = total effect; * *p* < 0.05; ** *p* < 0.01; *** *p* < 0.001.

**Figure 3 nutrients-14-02227-f003:**
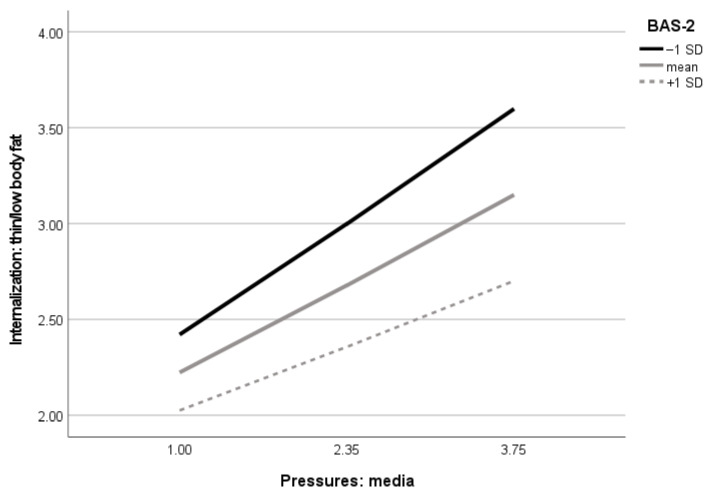
Visualization of body appreciation moderating effect in the association between media pressures and internalization of thin body with low fat in girls (*n* = 842). BAS-2 = Body Appreciation Scale-2, SD = standard deviation.

**Figure 4 nutrients-14-02227-f004:**
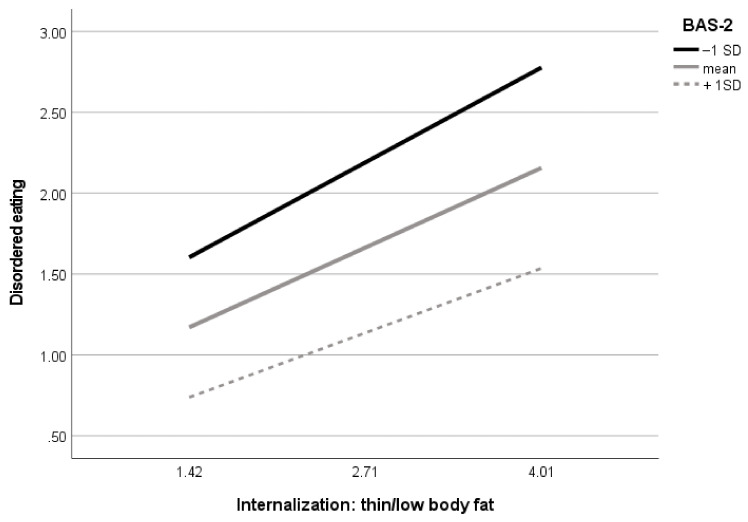
Visualization of body appreciation moderating effect in the association between internalization of thin body with low fat and disordered eating in girls (*n* = 842). BAS-2 = Body Appreciation Scale-2, SD = standard deviation.

**Figure 5 nutrients-14-02227-f005:**
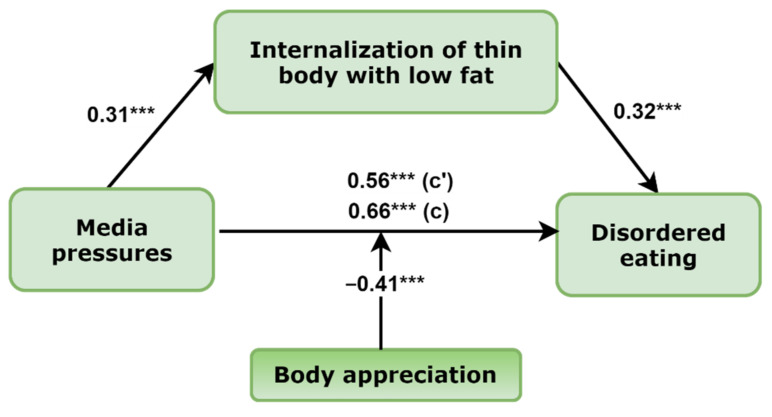
The final moderated mediation model in boys with the mediating effect of internalization of thin body (*n* = 570). Note: standardized path coefficients are presented; c’ = direct effect, c = total effect; *** *p* < 0.001.

**Figure 6 nutrients-14-02227-f006:**
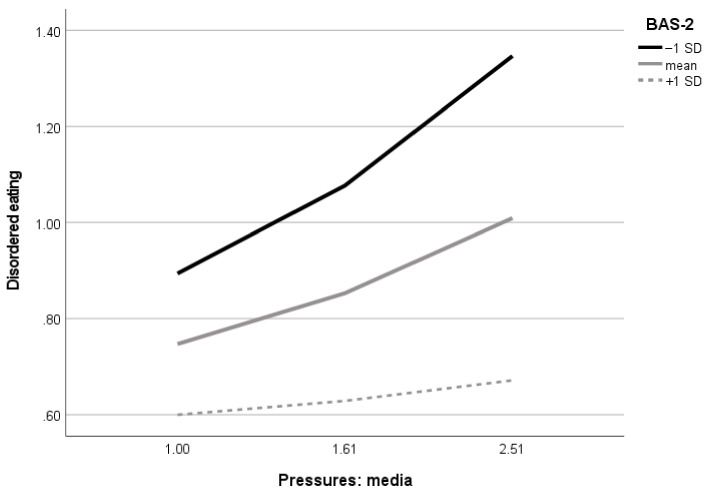
Visualization of body appreciation moderating effect in the association between perceived media pressures towards appearance and disordered eating in boys (*n* = 570). BAS-2 = Body Appreciation Scale-2, SD = standard deviation.

**Table 1 nutrients-14-02227-t001:** Comparison of the study variables (m ± SD) across boys’ and girls’ groups (*n* = 1412).

Variables	Boys, *n* = 570	Girls, *n* = 842	*t*	Cohen’s d	*p*
Age, years	17.02 ± 0.45	16.94 ± 0.47	3.2	2.18	0.001
Body mass index, kg/m^2^	21.99 ± 2.98	20.96 ± 3.06	6.3	0.34	<0.001
Time browsing internet, hours/day	3.80 ± 2.31	4.30 ± 2.29	−4.1	−0.22	<0.001
Media pressures	1.61 ± 0.90	2.35 ± 1.40	−12.1	−0.61	<0.001
Internalization of thin body	2.20 ± 1.10	2.71 ± 1.30	−8.0	−0.41	<0.001
Disordered eating	0.87 ± 0.87	1.72 ± 1.33	−14.4	0.76	<0.001
Body appreciation	3.41 ± 1.16	3.24 ± 1.13	2.6	0.15	0.009

m = mean, SD = standard deviation, *t* = *t*-test, *p* = significance level.

**Table 2 nutrients-14-02227-t002:** Correlations between study variables (*n* = 1412).

Variables	BMI	TI	MP	ITH	DE	BA
Body mass index, kg/m^2^ (BMI)	1	0.03	0.17 **	0.28 **	0.39 **	−0.28 **
Time browsing internet, hours/day (TI)	0.04	1	0.02	−0.007	0.08 *	−0.09 **
Media pressures (MP)	0.11 **	0.05	1	0.45 **	0.50 **	−0.29 **
Internalization of thin body (ITH)	0.22 **	0.09 *	0.31**	1	0.61 **	−0.34 **
Disordered eating (DE)	0.34 **	0.18 **	0.34 **	0.39 **	1	−0.57 **
Body appreciation (BA)	−0.13 **	−0.05	−0.16 **	0.006	−0.27	1

* *p* < 0.05; ** *p* < 0.01. Correlations between study variables for boys are presented in the lower diagonal while for girls in the upper.

## Data Availability

The dataset generated and analyzed during the current study is not publicly available but is available from the corresponding author on reasonable request.

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
