# Peer review of "Media Pressures, Internalization of Appearance Ideals and Disordered Eating among Adolescent Girls and Boys: Testing the Moderating Role of Body Appreciation"

_nutrients, 2022, doi:10.3390/nu14112227_

Round 1

Reviewer 1 Report

The present paper is aimed at exploring a possibile moderating role of body appreciation in a mediation model of media pressures, internalization of appearance ideals and disordered eating in a sample of 1412 adolescents (from the general population). The analyses were conducted separately in male and females. Results evidenced that, in the females group, body appreciation moderated associations between media pressures and internalization of thin body ideal and between internalization of thin appearance ideals and disordered eating. In the males group body appreciation moderated only the association between media pressures and disordered eating.

The study is well written and clearly conducted. The topic is interesting.

I have, therefore, only some minor point of concerns and suggestions that I explain below.

1- The study was entirely conducted before the emergence of the COVID-19 pandemic, making these results probably not perfectly extendable to the current period. I suggest authors specify better the period in which the survey was conducted (from what month to what month of 2019) and point out in the discussion section that the pandemic may have influenced some of these findings (future studies should be necessary).

2- The choice to conduct the analysis separately for males and females is understandable but not adequately reported in the introduction (as a prior hypothesis). I advise the authors to specify this point and to clarify better the rationale that led to this decision.

Furthermore, an analysis considering the whole sample (possibly reported in the supplementary materials) could be interesting.

3- I suggest to report the mean age of the two group in table 1.

4- An interesting finding is that males spend less time on the internet than females. Are there any hypotheses regarding this finding?

Author Response

Dear Reviewer,

Thank you for your time reviewing our paper and for your valuable comments. All changes made in the text are highlighted in a blue font.

The present paper is aimed at exploring a possible moderating role of body appreciation in a mediation model of media pressures, internalization of appearance ideals and disordered eating in a sample of 1412 adolescents (from the general population). The analyses were conducted separately in male and females. Results evidenced that, in the females group, body appreciation moderated associations between media pressures and internalization of thin body ideal and between internalization of thin appearance ideals and disordered eating. In the males group body appreciation moderated only the association between media pressures and disordered eating.

The study is well written and clearly conducted. The topic is interesting.

Thank you for the positive comment.

I have, therefore, only some minor point of concerns and suggestions that I explain below.

1- The study was entirely conducted before the emergence of the COVID-19 pandemic, making these results probably not perfectly extendable to the current period. I suggest authors specify better the period in which the survey was conducted (from what month to what month of 2019) and point out in the discussion section that the pandemic may have influenced some of these findings (future studies should be necessary).

 Thank you for this comment. The study was conducted before the pandemic and lockdown had no influence on the results.  We have specified the information about the period of the survey in months.

2- The choice to conduct the analysis separately for males and females is understandable but not adequately reported in the introduction (as a prior hypothesis). I advise the authors to specify this point and to clarify better the rationale that led to this decision.

We included explanation in the text why we decided to analyze girls and boys separately in Introduction prior the hypotheses.

Furthermore, an analysis considering the whole sample (possibly reported in the supplementary materials) could be interesting.

Thank you for this comment. Starting from Table 1 it could be seen that all study measures demonstrated significant differences across gender groups. Next, Table 2 confirms that correlations of the study measures were stronger in girls. The initial model (not presented in the manuscript) was not invariant across gender groups. In addition, as it is demonstrated in Figures 2 and 5, body appreciation had moderating effects on the association between media pressures and internalization of thin body as well as between internalization and disordered eating in girls. In boys, the direct effect from media pressures to disordered eating was moderated by body appreciation. Thus, for the total sample all three effects are moderated by body appreciation masking important differences in gender groups. That was the reason to analyze the data separately in boys and girls.

3- I suggest to report the mean age of the two group in table 1.

Mean age has been reported in Table 1.

4- An interesting finding is that males spend less time on the internet than females. Are there any hypotheses regarding this finding?

We included text in the Discussion explaining why adolescent boys spent less time using internet compared to girls.

Sincerely,

The authors Migle and Rasa

Reviewer 2 Report

Thank you for the opportunity to review this manuscript. The results are very interesting.  I only have a few minor comments:

  1. In the theoretical introduction, there is no more detailed justification why the model is tested separately in boys and girls.
  2. Inconsistency in the description of the results in the abstract and in the method (e.g. average age of girls - 16.9 vs.16; percentage of boys - 40.2 vs 40.1).
  3. In discussion, there is no more detailed explanation as to why the results for boys and girls were different.

Author Response

Dear Reviewer,

Thank you for your time reviewing our paper and for your valuable comments. All changes made in the text are highlighted in a blue font.

Thank you for the opportunity to review this manuscript. The results are very interesting.  I only have a few minor comments:

1. In the theoretical introduction, there is no more detailed justification why the model is tested separately in boys and girls.

Thank you for your comment. The explanation was included to the Introduction.

2. Inconsistency in the description of the results in the abstract and in the method (e.g. average age of girls - 16.9 vs.16; percentage of boys - 40.2 vs 40.1).

Thank you for the remark, we corrected typos in the text.

3. In discussion, there is no more detailed explanation as to why the results for boys and girls were different.

We included short explanation in the Discussion about possible differences findings of boys and girls.

Sincerely,

The authors Migle and Rasa